# Clinical Evaluation of a New Spectral-Domain Optical Coherence Tomography-Based Biometer

**DOI:** 10.3390/diagnostics14050560

**Published:** 2024-03-06

**Authors:** Jorge L. Alió, Marina José-Martínez, Antonio Martínez-Abad, Alejandra E. Rodríguez, Francesco Versaci, Jesper Hjortdal, Joaquim Neto Murta, Ana B. Plaza-Puche, Mario Cantó-Cerdán, David P. Piñero

**Affiliations:** 1Research and Development Department, Vissum Grupo Miranza, 03016 Alicante, Spain; jlalio@vissum.com (J.L.A.); marina.jose@vissum.com (M.J.-M.); anmartinez@vissum.com (A.M.-A.); abplaza@vissum.com (A.B.P.-P.); mcanto@vissum.com (M.C.-C.); 2School of Medicine, Miguel Hernandez University, 03202 Alicante, Spain; 3CSO (Construzioni Strumenti Oftalmici), 50018 Scandicci, Italy; f.versaci@csoitalia.it; 4Department of Opthalmology, Aarhus University Hospital, 8000 Aarhus, Denmark; jesper.hjortdal@clin.au.dk; 5Department of Opthalmology, University of Coimbra, 3000-548 Coimbra, Portugal; jmurta@outlook.pt; 6Department of Optics, Pharmacology and Anatomy, University of Alicante, 03690 Alicante, Spain; david.pinyero@gcloud.ua.es

**Keywords:** optical biometry, axial length, anterior chamber depth, central corneal thickness, swept-source optical coherence tomography, spectral-domain optical coherence tomography

## Abstract

The VEMoS-AXL system is a new optical biometer based on spectral domain optical coherence tomography (SD-OCT) that has been tested in terms of intrasession repeatability and compared with a swept-source optical coherence tomography biometer (SS-OCT), which is recognized as the gold standard for the performance of an agreement analysis. A biometric analysis was performed three consecutive times in 120 healthy eyes of 120 patients aged between 18 and 40 years with the SD-OCT system, and afterwards, a single measurement was obtained with the SS-OCT system. Within-subject standard deviations were 0.004 mm, 4.394 µm, and 0.017 mm for axial length (AL), central corneal thickness (CCT), and anterior chamber depth (ACD) measures obtained with the SD-OCT biometer, respectively. The agreement between devices was good for AL (limits of agreement, LoA: −0.04 to 0.03 mm) and CCT (LoA: −4.36 to 14.38 µm), whereas differences between devices were clinically relevant for ACD (LoA: 0.03 to 0.21 mm). In conclusion, the VEMoS-AXL system provides consistent measures of anatomical parameters, being most of them interchangeable with those provided by the SS-OCT-based gold standard.

## 1. Introduction

Several advances have been developed and introduced in clinical practice to improve the refractive predictability of cataract surgery, such as highly optimized intraocular lens (IOL) power calculation formulas and very accurate optical biometry devices [1]. Specifically, advances in optical biometry have allowed clinicians to obtain very precise measurements of different ocular dimensions in a fast and non-invasive mode [2]. One of the most commonly used optical biometry systems still considered as a gold standard for the determination of axial length (AL) [3] in research and clinical practice is the IOL-Master biometer from Carl Zeiss Meditec (Jena, Germany). This biometer, which is based on partial coherence interferometry (PCI), has demonstrated to be reliable for measuring AL as well as other ocular dimensions [4,5,6], these being data useful to obtain accurate IOL power calculations [7]. The last version of this technology (IOL-Master 700 (Carl Zeiss Meditec AG, Jena, Germany)) has been implemented with swept-source optical coherence tomography (SS-OCT) to provide a more comprehensive analysis of the anterior segment and consequently to optimize IOL power calculations [8]. Specifically, it automatically takes swept source OCT scans to determine several parameters, such as AL, lens thickness, or anterior chamber depth (ACD). Specifically, AL measurements are the average values of three scans in each of six meridians [5]. The reliability of this updated biometer has also been investigated and confirmed in previous studies [9,10].

Besides the evident advance in optical biometry in recent years, new systems are continuously being developed using different optical principles which must be clinically validated. Considering the IOL Master platform as the gold standard [3], as previously mentioned, most studies investigating the reliability of new optical biometers are based on comparisons with such platforms [4,5,6,10,11,12,13]. Recently, a new optical biometer, VEMoS-AXL (CSO, Firenze, Italy), based on spectral-domain OCT (SD-OCT) has been developed. The purpose of this study was to investigate the reliability of the biometric measurements provided by this new system and analyze the agreement with those obtained with the gold standard IOL-Master 700. It should be considered that a clinical evaluation of a new device commercially released for clinical purposes is required to ensure that it can be used in clinical practice to take correct decisions. The only difference between the new biometer evaluated in the current study and the gold standard is the optical principle used, which provides a better axial resolution and can be integrated into OCT-based equipment for evaluating other aspects of the eye. After investigating the clinical usefulness of the new system, future studies should be conducted to investigate in depth the potential specific benefits of this new technology with regard to its clinical relevance.

## 2. Materials and Methods

This observational, prospective study included 120 eyes of 120 patients aged between 18 and 40 years. Eyes were selected randomly; a random number sequence (dichotomic sequence 0 and 1) was designed using a computer software. Patients were selected from the consultation of Vissum Alicante, Grupo Miranza. Patients were informed about the purpose of the study and signed the informed consent form. The study complied with the tenets of the Declaration of Helsinki, and it was approved by the Ethics Committee of Comité Ético de Investigación Médica Instituto de Microcirugía Ocular from Spain with reference 210318-169. The inclusion criterion were patients from 18 to 40 years of age and with less than 2D of astigmatism. The exclusion criterion were ocular pathology affecting visual capabilities, reduced transparency of ocular media, and systemic disease affecting visual capabilities.

A complete ophthalmologic examination was carried out to all patients to determine the ones to include in the study. This examination included the IOLMaster 700 (Carl Zeiss Meditec AG, Jena, Germany) and the VEMoS-AXL (Costruzione Strumenti Oftalmici (CSO), Scandicci FI, Italy). The same experienced independent examiners performed all measurements.

### 2.1. Swept-Source Optical Coherence Tomography Biometer

The IOL-Master 700 is a computerized biometry device used to measure distances in the human eye along the visual axis. The device acquires multiple measurements for each of the various eye parameters in 1 measurement-capture process and presents an average value measurement.

The length measurement is based on swept-source frequency-domain OCT enabling a 44 mm scan depth with 22 μm resolution in tissue. The speed of the length measurement system allows the acquisition of full-eye length tomograms at 2000 A-scans/s. On the contrary to the PCI biometry and all other optical biometry devices from various manufacturers using optical A-scans, swept-source biometry applies optical B-scan technology to determine the biometric data. The optical B-scan technology allows cross-sectional visualization of structures along the visual axis. Thus, the examiner can check whether or not ocular interfaces are detected correctly by the algorithm [8].

### 2.2. VEMoS-AXL Topographer and Biometer

The VEMoS-AXL prototype (running on Phoenix software version 4.1, CSO, Scandicci FI, Italy) is an evolution of the Anterior Segment OCT MS-39. Like its predecessor, it uses SD-OCT and Placido-disk corneal topography to obtain measurements of the anterior segment of the eye. The VEMoS AXL acquires 1 keratoscopy image with 22 rings and a series of 25 SD-OCT radial scans at a wavelength of 845 nm, with an axial resolution of 3.6 μm and a maximum depth of 7.5 mm. Each scan is 16 mm long and includes 1024 A-scans. The ring edges are detected on the Placido image so that elevation, slope, and curvature data of the anterior corneal surface can be measured using an arc-step method with conic curves. Only profiles of the posterior cornea, anterior lens, and iris are derived from the SD-OCT scans. Data for the anterior surface from the Placido image and SD-OCT scans are merged using a proprietary method.

Unlike the MS-39, VEMoS-AXL can perform biometric measurements. During the acquisition, the operator can see both a frontal picture of the eye and a sectional picture of the anterior segment in order to ensure the optimal alignment of the patient eye. The output of this live modality consists of a keratoscopy and 6 pairs of sections (one comprising the cornea and the second one comprising the retinal layers) the distance is measured by a micrometric encoder. The axial length can be determined, therefore, by the position of corneal vertex on the first image, the position of retinal pigmented epithelium (RPE) in the retinal image and the relative optical distance between the two images accounted by the micrometric encoder.

### 2.3. Statistical Analysis

Statistical analysis of data was performed using the SPSS software for Windows (version 11.0, SPSS Inc., Chicago, IL, USA Statistical Package for Social Sciences). The normality of all data distributions was confirmed by the Kolmogorov–Smirnov test, and parametric statistics were always applied. The unpaired Student t test was used for comparing the instruments in each parameter. Pearson coefficient was calculated to assess the correlation between parameters measured with each device. All tests were two-tailed, and a *p* value less than 0.05 was considered statistically significant. The statistical power of the sample has been calculated by GRANMO program (version 7.11, online) obtaining a power of 99.9.

Intrasession repeatability of the measurement method was analyzed using the intraclass correlation coefficient (ICC) and the within-subject standard deviation (S_w_). The Bland–Altman analysis was used for studying the interchangeability of both devices to measure the anatomical parameters. This method shows the agreement between two clinical procedures. Bland–Altman plots show the differences between the methods plotted against the mean of the 2 methods. The limits of agreement (LoA) are defined as the mean ±1.96 × SD of the differences. If the limits are clinically relevant, the 2 methods cannot be used interchangeably.

## 3. Results

One hundred and twenty patients with a mean age of 24.5 ± 5.2 years old, comprising 82 females (68.3%) and 38 males (31.7%), were enrolled in this study. No statistically significant differences were observed between gender-based groups (*p* > 0.05).

### 3.1. Intrasession Repeatability Analysis

The intraclass correlation coefficient for the six consecutive measurements of AL performed by VEMoS-AXL was 0.994 (*p* < 0.001) and the within-subject standard deviation was 0.02. The mean descriptive data provided by VEMoS-AXL are shown in Table 1.

### 3.2. Analysis of the Agreement between Biometers

The comparison of AL measured with VEMoS-AXL and IOL-Master did not show significant differences, with mean values of 23.83 ± 1.01 mm and 23.82 ± 1.00 mm, respectively (*p* = 0.271). The correlation of AL measures with both devices was very strong (r = 0.994, *p* < 0.001). For CCT measurements, statistically significant differences were found between both devices, with mean values of 545.86 ± 30.92 µm and 540.85 ± 32.51 µm with VEMoS-AXL and IOL-Master biometers, respectively (*p* < 0.001). Despite the statistical disparities observed in the differences found, the correlation between both devices was strong (r = 0.990, *p* < 0.001). Statistically significant differences were also found between the ACD measurements obtained with the two biometers evaluated, with mean values of 3.70 ± 0.28 mm and 3.57 ± 0.29 mm for VEMoS-AXL and IOL-Master, respectively (*p* < 0.001). Furthermore, the correlation of ACD measures obtained with both biometers was also found to be strong and statistically significant (r = 0.996, *p* < 0.001).

The agreement of all biometric parameters measured with the two biometers evaluated is graphically displayed in Figure 1 (Figure 1A—axial length; Figure 1B—central corneal thickness; Figure 1C—anterior chamber thickness) and numerically reported in Table 2.

Finally, the correlation of the difference obtained in the measures of the variables analyzed between both devices and the magnitude of such variables was investigated to check if the discrepancy between biometers increased for longer and shorter eyes as well as for thicker and thinner corneas. No correlation of the differences between devices in ACD (r = −0.011, *p* = 0.906) and AL (r = 0.163, *p* = 0.076) with the magnitudes of such variables was found. On the contrary, a weak but statistically significant correlation of the pachymetric differences between devices and the magnitude of CCT was obtained (r = −0.266, *p* = 0.003).

## 4. Discussion

Several studies have investigated the level of intrasession repeatability of the last generation of optical biometers, including those based on SS-OCT [14,15,16,17,18], in order to know the consistency of the anatomical measures obtained with these devices. It should be considered that such measurements are crucial in the clinical practice, especially for IOL power calculations when planning cataract surgeries. Recently, a new optical biometer (VEMoS-AXL system) based on a different optical principle, which is SD-OCT, has been developed and presented as a potentially useful device in clinical practice. However, to this date, there are no studies confirming the consistency of the measurements obtained by this new device and if the measurements that provide are comparable to those obtained with the already commercially available biometers. This type of research is crucial in clinical research to confirm if new devices commercially developed and released can be used with accuracy for taking decisions in the real clinical practice. In the current study, an evaluation of intrasession repeatability of the measures obtained with the VEMoS-AXL system was performed first, and afterwards, a comparison of such measures with those obtained using a gold standard in terms of optical biometry, the IOL-Master 700 system.

The consistency of AL, CCT, and ACD measures obtained for the new optical biometer evaluated was excellent in the sample of healthy eyes evaluated, with S_w_ of 0.004 mm, 4.394 µm, and 0.017 mm, respectively. These S_w_ values were similar to those reported for the anatomical measures obtained with PCI, SS-OCT, optical low-coherence reflectometry (OLCR), and Scheimpflug camera-based biometers [4,14,15,16,17,18]. Hashemi et al. [14] reported S_w_ values of 0.02 mm and 0.02 mm for AL and ACD measurements, respectively, obtained with an instrument consisting of a dual rotating Scheimpflug camera, a Placido disk topographer, and an optical coherence tomography-based A-scan. Monera Lucas et al. [15] obtained S_w_ values of 0.005 mm, 0.043 mm, and 1.1 µm when evaluating the intrasession repeatability of the measurements of AL, ACD, and CCT, respectively, obtained with a SS-OCT biometer. With this same SS-OCT-based device, Cheng et al. [17] reported mean S_w_ values of 0.0064 mm, 0.011 mm, and 1.25 µm for AL, ACD, and CCT measures, respectively. Fisus et al. [18] compared the level of intrasession repeatability of three optical biometers, two SS-OCT-based devices (IOL-Master 700 from Carl Zeiss Meditec and Anterion from Heidelberg Engineering (Heidelberg, Germany)) and one biometer based on OLCR (Lenstar LS900 from Haag-Streit). These authors found S_w_ values of 0.006, 0.008, and 0.012 mm for the measurements of AL obtained with IOL-Master, Anterion, and Lenstar systems, respectively [18]. Regarding ACD, these authors reported mean S_W_ values of 0.039, 0.004, and 0.134 mm, respectively [18]. Finally, the mean S_W_ associated with CCT measures was 5.556, 5.423, and 3.834 µm for IOL-Master, Anterion, and Lenstar systems, respectively [18]. Therefore, the intrasession repeatability obtained with the new biometer evaluated in the current series is within the range reported for other currently available biometers, confirming that this device can provide consistent measures of AL, ACD, and CCT in healthy eyes. Future studies should be conducted to confirm if this trend is also observed in pathological eyes or eyes with previous ocular surgery.

Besides the analysis of consistency of the measurements obtained with the VEMoS-AXL system, an analysis of agreement with the gold standard IOL-Master 700 from Carl Zeiss Meditec was also performed. Regarding AL, the mean difference between both devices was −0.01 mm, and such a difference did not reach statistical significance. Likewise, the difference was not clinically relevant, considering that the limits of agreement between biometers was of −0.04 mm and 0.03 mm, values consistent with those reported in comparative studies, concluding that the compared biometers were interchangeable (limits of agreement between −0.09 to 0.09 mm) [6,13,15,16,17]. It should be considered that a 0.1 mm error in AL is equivalent to an error of about 0.27 D in the spectacle plane assuming normal eye dimensions, a dioptric value that can be detected and measured in subjective refraction [19]. Moon et al. [16] compared three different biometers (Anterion vs. IOL-Master 500 vs. OA-2000), analyzing the level of interchangeability between the anatomical measures obtained with them. These authors found limits of agreement between pairs of biometers that could be considered as clinically acceptable: −0.09 to 0.08 mm for Anterion vs. OA-2000, −0.04 to 0.09 mm for Anterion vs. IOL-Master 500, and −0.01 to 0.07 mm for OA-2000 vs. IOL-Master 500 [16]. Monera Lucas et al. [15] confirmed the interchangeability of AL measures from Anterion and Lenstar LS900 systems, obtaining limits of agreement of −0.06 and 0.04 mm. On the contrary, Kim et al. [5] concluded that a SS-OCT biometry system could not be used interchangeably with a PCI biometer when measuring AL, as the limits of agreement were −0.15 and 0.21 mm.

The agreement of the ACD measures obtained with the VEMoS-AXL and IOL-Master 700 biometers was also investigated. The mean difference between biometers was 0.12 mm, with limits of agreement of 0.03 and 0.21 mm. This difference was statistically significant as well as clinically relevant, considering the trend an overestimation of ACD with the VEMoS-AXL system and that the limits of agreement were in the range of those reported in studies comparing biometers and concluding that ACD measures were not interchangeable [5,6,13,14,15,16,17]. Moon et al. [16] concluded that ACD measures obtained with Anterion and IOL-Master 500 systems were not interchangeable (limits of agreement, −0.33 to 0.22 mm) as well as ACD measurements obtained with OA-2000 and IOL-Master 500 biometers (limits of agreement, −0.37 to 0.20 mm). Cheng et al. [17] compared different biometers, confirming the presence of a poor agreement of ACD measures obtained with some of them: Anterion vs. OA-2000 (−0.19 to 0.28 mm), OA-2000 vs. IOL-Master (−0.19 to 0.25 mm), Anterion vs. Lenstar (−0.04 to 0.21 mm), and OA-2000 vs. Lenstar (−0.16 to 0.23 mm). Likewise, the comparison between ACD measurements obtained with Pentacam AXL and IOL-Master biometers also revealed the presence of clinically significant differences between devices (limits of agreement, −0.246 to 0.345 mm). In the current study, this trend of the SD-OCT-based biometer to provide, on average, 120 µm-longer values of ACD compared to the SS-OCT-based biometer may be attributed to several factors that should be investigated further, such as differences in the level of resolution, the reference plane used for the measurements, or in the algorithm used to delimitate the edges of the structures at the corneal and retinal plane. A similar trend was also reported in a previous study comparing the ACD measures obtained with a SD-OCT (Catalys) and a SS-OCT (Anterion) system, obtaining mean values of ACD in the sample evaluated of 3.33 ± 0.37 mm and 3.15 ± 0.37 mm, respectively [20]. The authors attributed this difference to the lower resolution of the SD-OCT system (<30 μm) compared to the SS-OCT used (<10 µm).

Finally, an agreement analysis of the pachymetric readings obtained with the two biometry systems compared in the current study was also performed. Mean difference in CCT between VEMoS-AXL and IOL-Master 700 systems was 5.01 µm, with limits of agreement of −4.36 to 14.38 µm. Although the difference was found to be statistically significant, with a minimal trend of VEMoS-AXL system to provide larger CCT values, the limits of agreement could be considered as acceptable from a clinical perspective. It should be considered that potential pachymetric errors until approximately 14 µm do not have any relevant impact on clinical decisions. This value on average represents 2.6% of mean value of CCT measured with the SD-OCT-based system. It should be considered that pathological or post-surgical changes in CCT of more than 3% can have a significant impact on visual quality [21]. Cheng et al. [17] confirmed in their comparative study that CCT measures were not interchangeable between some of the devices tested: Anterion vs. OA-2000 (−22.83 to −0.99 µm), OA-2000 vs. IOL-Master (−24.37 to −2.70 µm), and OA2000 vs. Lenstar (−26.88 to −3.94 µm). A limitation of the present study is related to the characteristics of the sample using only young subjects without cataracts. Future studies should compare and validate the results in these cases where lens transparency is altered.

## 5. Conclusions

The new optical biometry system VEMoS-AXL from CSO provides consistent measures of AL, ACD, and CCT in healthy eyes. AL and pachymetric measurements can be used interchangeably with those obtained with the SS-OCT-based system IOL-Master 700, which is considered the gold standard in optical biometry. This confirms the viability of the clinical use of this new biometer. Special care should be taken with ACD measures of both devices that cannot be used interchangeably, which deserves more research. Future studies should be also conducted to confirm all these outcomes in eyes with different pathological conditions and after different types of ocular surgeries.

## Figures and Tables

**Figure 1 diagnostics-14-00560-f001:**
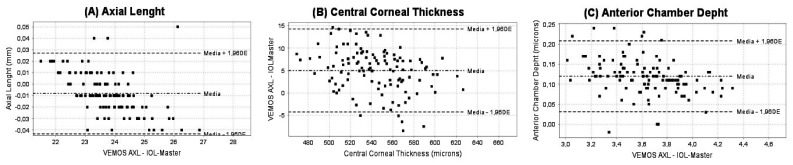
Bland–Altman plots of the agreement of the measurement of axial length (**A**), central corneal thickness (**B**), and anterior chamber depth (**C**) obtained with the optical biometers VEMoS-AXL and IOL-Master 700. The dashed lines show the lower and upper limits of agreement and the dotted middle line the shows the difference between measurement techniques.

**Table 1 diagnostics-14-00560-t001:** Mean biometric parameters and their corresponding standard deviation obtained by VEMOS-AXL device (S_W_: within-subject standard deviation; mm, millimeters; µm: micrometers; AL, axial length; CCT, central corneal thickness; ACD, anterior chamber depth).

	Mean	SD	Minimum	Maximum
AL (mm)	23.83	1.01	21.49	26.88
S_W_ of AL (mm)	0.004	0.006	0.000	0.003
CCT (µm)	545.86	30.92	473.70	626.73
S_W_ of CCT (µm)	4.394	1.531	0.000	7.140
ACD (mm)	3.695	0.279	3.090	4.360
S_W_ of ACD (mm)	0.017	0.027	0.000	0.250

**Table 2 diagnostics-14-00560-t002:** Agreement between VEMoS-AXL and IOL-Master devices for axial length (AL), central corneal thickness (CCT) and anterior chamber depth (ACD). (MD: mean difference; SD: standard deviation; mm, millimeters; μm: microns).

Mean ± SD	VEMoS-AXL	IOL-Master	*p*-Value	Differences VEMoS-IOL-Master			
				Mean	SD	Minimum	Maximum
AL (mm)	23.83 ± 1.01	23.82 ± 1.00	0.271	0.01	0.02	−0.07	0.07
CCT (µm)	545.86 ± 30.92	540.85 ± 32.51	<0.001	5.01	4.73	−8.48	14.65
ACD (mm)	3.70 ± 0.28	3.57 ± 0.29	<0.001	0.12	0.05	−0.02	0.24

## Data Availability

Data are available upon reasonable request.

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
