# Peer review of "Clinical Evaluation of a New Spectral-Domain Optical Coherence Tomography-Based Biometer"

_diagnostics, 2024, doi:10.3390/diagnostics14050560_

Round 1
Reviewer 1 Report
Comments and Suggestions for Authors
The submitted manuscript by Ailo et al. reported the study results comparing the measurement agreement of the eye biometry data, including the axial eye length, central corneal thickness, and the anterior chamber distance collected using a new SD-OCT (VEMoS AXL system) and the existing SS-OCT based biometry instrument (IOL master). The reviewer generally found the submitted manuscript well written with sufficient detailed information to ensure the study comparison is fair and not biased. However, the reviewer still has some questions or comments raised while reviewing the manuscript that would like the authors to clarify further. Please see the comments listed below.
1. In sections 2.1 and 2.2, the authors provided the specifications for the SS-OCT and SD-OCT-based biometry instruments used in this study. The authors should also consider summarizing the difference in the specification between these two instruments in a tabulated manner. This should allow the readers to understand the difference between the two instruments quickly. It would be helpful if the authors could provide the measurement time information and the A-scan rate for the SD-OCT barometer in this table.
2. Based on the information provided by the authors, the SD-OCT-based biometry instrument includes two different modules, which use SD-OCT and Placido-88 disk corneal topography to obtain measurements of the anterior segment of the eye. Then, a separate measurement of sections showing the retinal layers was performed. Finally, the axial eye length was acquired by measuring the shift between the section of the anterior segment, and the section of the retinal layers was computed afterward. Since the optical and optomechanical design to allow high-quality retinal OCT imaging is not the same as the one used to acquire anterior segment images, it would be helpful if the authors could comment on how the section of the retinal layers was acquired. Also, please comment on the time required to perform one full measurement, and how it compared to the IOL master.
3. Based on the results demonstrated in this submitted manuscript, it seems the SD-OCT-based biometry instrument exhibits a comparable measurement performance with the IOL master one. However, as we know, in order to promote this new instrument for biometry application, other parameters, including the pricing, might matter as well. It would be helpful if the authors could comment on this part as well.
Reviewer 2 Report
Comments and Suggestions for Authors
The authors compare measurements between the VEMoS AXL and an IOL Master, with the last one as the standard. The current paper structure looks like a white paper testing medical equipment. It requires further work to have a full research paper with all the required information for a specialized and general audience.
1. A common situation in the manuscript is the introduction of acronyms without previous definitions since the abstract, i.e., SPSS, AMA, VEMoS, etc.
2. Title. “validation…” It is unclear how the authors validate the system in the current report, as they only compare measurements. The latter gives an idea about the devices’s performance, but the reviewer is unsure how this validates it.
3. Abstract. The sentence that includes: for agreement analysis is confusing. The reviewer is not sure of the purpose of this information or why it is considered the gold standard with such an argument.
4. The abstract is saturated with non-defined or non-introduced figures. It is confusing to understand the focus of the work when the last part indicates “consistent measurements…”. This does not match the title’s term validation.
5. Page 2, “…examiners (AMA and MCC),” what does it mean?
6. Page 3. The merge of the Placido images with the SD-OCT is omitted and never shown. Understandably, the algorithm will not be fully described, but there is a lack of information on this matter. Besides, the device is within the range of the current biometers. Is this feature an advantage to claim or not? Otherwise, what is the point of having a new system with the same performance? Where are the advantages and advances in the field?
7. Page 3. The MS-39 model is mentioned in the text, but there is no table to compare the hardware parameters of the biometers. How better is the new one according to these specs? Where is the advance?
8. Page 3. The value 1.96SD looks wrong.
9. Page 3. Why are sentences highlighted in yellow?
10. Page 4. The first paragraph mentions a comparison; however, the reviewer could not find the table showing this information. It is necessary to see it and not just the texted summary.
11. Page 4. Please check sentences like “The statistical difference of differences” … is confusing. Or typos as Figure 1 caption “…dotted line the shows the…”.
12. Page 4. Figure 1 is too small, and it is impossible to see the information or dotted lines in it. Please improve this as it should be relevant.
13. Page 5. The authors indicate that the system’s repeatability is within the range of reported biometers. Why is this a surprise? It is based on optical coherence tomography (OCT), detecting layer interfaces within a low-scattering media like the eye. Temporal, swept source, spectral (Fourier domain), and maybe in the future, with higher pair photon sources, quantum OCT can detect these layers. The system’s novelty is not apparent at this point; however, as it adds the Placido rings results, this may give more information than a regular biometer. However, this is not clearly stated in the manuscript; page 5 is full of figures that are hard to follow since the target and focus of the paper were not specified.
14. Conclusions. The repeatability and consistency of a new clinical device are performed for the company that introduces it and a group of researchers that proves it. The reviewer is unsure how this process could be presented as research when information on the system, how it works, and how it improves current devices is missing in the current manuscript. Simple equipment testing with results within the range of commercial devices is not enough for an original research article.
Comments on the Quality of English LanguageAlready included in the authors' suggestions.
